# RealRAG: Retrieval-augmented Realistic Image Generation via Self-reflective Contrastive Learning

Yuanhuiyi Lyu [1]  Xu Zheng [1]  Lutao Jiang [1]  Yibo Yan [1]  Xin Zou [1]  Huiyu Zhou [2]
Linfeng Zhang [3]  Xuming Hu [1 2 4]

## Abstract

Recent text-to-image generative models, *e.g.*, Stable Diffusion V3 and Flux, have achieved notable progress. However, these models are strongly restricted to their limited knowledge, *a.k.a.*, their own fixed parameters, that are trained with closed datasets. This leads to significant hallucinations or distortions when facing fine-grained and unseen novel real-world objects, *e.g.*, the appearance of the Tesla Cybertruck. To this end, we present **the first** real-object-based retrieval-augmented generation framework (**RealRAG**), which augments fine-grained and unseen novel object generation by learning and retrieving real-world images to overcome the knowledge gaps of generative models. Specifically, to integrate missing memory for unseen novel object generation, we train a reflective retriever by **self-reflective contrastive learning**, which injects the generator's knowledge into the sef-reflective negatives, ensuring that the retrieved augmented images compensate for the model's missing knowledge. Furthermore, the real-object-based framework integrates fine-grained visual knowledge for the generative models, tackling the distortion problem and improving the realism for fine-grained object generation. Our Real-RAG is superior in its modular application to **all types** of state-of-the-art text-to-image generative models and also delivers **remarkable** performance boosts with all of them, such as a **gain of *16.18%*** FID score with the auto-regressive model on the Stanford Car benchmark.

[1]The Hong Kong University of Science and Technology (Guangzhou) [2]Guangxi Zhuang Autonomous Region Big Data Research Institute [3]Shanghai Jiao Tong University [4]The Hong Kong University of Science and Technology. Correspondence to: Xuming Hu <xuminghu@hkust-gz.edu.cn>.

*Proceedings of the 42nd International Conference on Machine Learning*, Vancouver, Canada. PMLR 267, 2025. Copyright 2025 by the author(s).

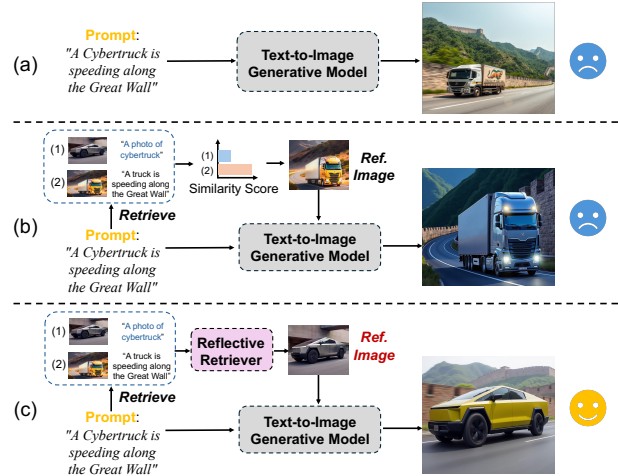

*Figure 1.* **(a)** The pipeline of text-to-image generative models. **(b)** The framework of existing retrieval-augmented methods. **(c)** The framework of our proposed RealRAG.

## 1. Introduction

Recent text-to-image generators have achieved notable progress in image synthesis from the given textual prompts. There are three mainstream types of generative models, including the U-Net-based diffusion model (Rombach et al., 2022a; Podell et al., 2023), the DiT-based diffusion model (Xiao et al., 2024; Sun et al., 2024), and the auto-regressive model (Esser et al., 2024; BlackForest, 2024). Typically, these models store all their visual memory (*e.g.*, the appearance of Big Ben) implicitly in the parameters of the underlying neural network, requiring a lot of parameters(*e.g.*, 10B). Furthermore, similar to the hallucination problem of Large Language Models (LLMs) (OpenAI, 2023; Touvron et al., 2023), the large-scale text-to-image generative models also show the same problem. Some generated images include ghosting, distortions, and unnatural elements when generating specific real-world objects. Therefore, these problems motivate the development of text-to-image generation models, which can integrate external visual knowledge (*e.g.*, images from the web) to augment generative realism and accuracy for fine-grained and unseen novel object generation.

Retrieval-augmented generation (RAG) has shown promise in natural language processing (NLP) (Gao et al., 2023). To enhance the specific knowledge and minimize the hallucination of LLMs, a retrieval model first retrieves the most relevant documents for the input prompts (*e.g.*, the questions), and then LLMs generate predictions powered by the recalled documents. However, in text-to-image generation, these RAG methods, which rely primarily on similarity matching, remain challenging. Specifically, as shown in Fig. 1 (**b**) and (**c**), the candidate image (**1**), with the highest similarity score of the text prompt, fails to improve the image generation. In contrast, candidate image (**2**), despite having a lower similarity score, complements the missing knowledge and augments the image generation. In this work, we propose a reflective retriever trained via self-reflective contrastive learning, **aimed at retrieving images with missing knowledge instead of the most relevant one**.

In this paper, we present the **first** real-object-based retrieval-augmented generation framework (RealRAG), which leverages real-world images to compensate for the missing knowledge inherent in generative models and improve realistic image generation. *The core insight of RealRAG is to enhance realism and reduce hallucination in text-to-image generative models powered by the reflective retriever, which is trained via self-reflective contrastive learning.* In addition, our proposed RealRAG demonstrates **remarkable flexibility** and can be applied across diverse text-to-image generative models, yielding significant performance gains, *e.g.*, a **+16.18% gain** with Emu (Sun et al., 2024) on the Stanford Cars benchmark (Krause et al., 2013).

Specifically, we **first** generate images from the given text prompts using the specific text-to-image generative models. Given these generated images, we then sample the reflective negatives in the image database by selecting images with the highest cosine similarity of the generated images. In this way, these sampled negatives are similar to the generated images that store the generative models' visual memory. **Second**, we train the *reflective retriever* with self-reflective contrastive learning, which utilizes text prompts as positives and trains with the sampled reflective negatives. In this way, the well-trained retriever can recall the prompt-relevant images also with missing knowledge for the generative models. For example, as shown in Fig. 1, our RealRAG retrieves the image ["Cybertruck"], instead of the image with highest similarity (["A truck is speeding along the Great Wall"]), which integrates missing knowledge for generative models.

We apply our **RealRAG** to *all types of state-of-the-art text-to-image generative models*, including U-Net-based diffusion models (SD V2.1 (Rombach et al., 2022a), SD XL (Podell et al., 2023)), DiT-based diffusion models (SD V3 (Esser et al., 2024), Flux (BlackForest, 2024)), and auto-

regressive models (OmniGen (Xiao et al., 2024), Emu (Sun et al., 2024)). Note that, our RealRAG is the first work to build **a unified RAG framework for all types of text-to-image generative models**. Our RealRAG consistently delivers significant performance improvements with all these models, such as **+5.77%** on the Stanford Cars benchmark with Flux (BlackForest, 2024) and **+20.48%** on the Oxford Flowers benchmark with Emu (Sun et al., 2024). Additionally, to evaluate the capability of unseen novel object generation, we collect recent novel objects from news web pages and construct human evaluations. Project page: `https://qc-ly.github.io/RealRAG-page/`

## 2. Related Work

### 2.1. Text-to-Image Generation

The U-Net-based Stable Diffusion models (Rombach et al., 2022a; Podell et al., 2023) first perform universal image generation from text prompts, which typically trained on large scale text-image paired dataset, *a.k.a.*, LAION 5B (Schuhmann et al., 2022). After the proposal of the Diffusion Transformer (DiT) (Peebles & Xie, 2023), some research, such as Stable Diffusion V3 (Esser et al., 2024) and Flux (BlackForest, 2024; Liu et al., 2025), utilize DiT as the backbone to develop DiT-based Diffusion Models for text-to-image generation. Recently, with the success of auto-regressive (AR) modeling in natural language processing (OpenAI, 2023; Touvron et al., 2023), some works have explored how to combine auto-regressive models with diffusion models to improve the understanding capability and further build a unified multi-modal model for both understanding and generation (Chen et al., 2024; Xie et al., 2024; Zhou et al., 2024). These AR-based models, such as OmniGen (Xiao et al., 2024) and Emu (Sun et al., 2024), also show notable performance on the text-to-image task. While these methods have achieved strong performance in text-to-image generation, they store all the knowledge in their pre-trained parameters, which leads to hallucinations and distortions when generating realistic objects. To address this limitation, we propose the real-object-based RAG framework to integrate missing knowledge and improve the ability to generate realistic images.

### 2.2. Retrieval-augmented Generation

Retrieval-augmented generation has shown promise with NLP (Lewis et al., 2020; Guu et al., 2020). To incorporate external knowledge into a LLM (Gao et al., 2023; Jiang et al., 2023b), these methods retrieve documents relevant to inputs from an external database, subsequently, the LLM utilizes the recalled documents as references to generate accurate results. The external knowledge used is typically a text database (Hashimoto et al., 2018; Khandelwal et al., 2019; Shi et al., 2023; Lyu et al., 2024b). However, the text

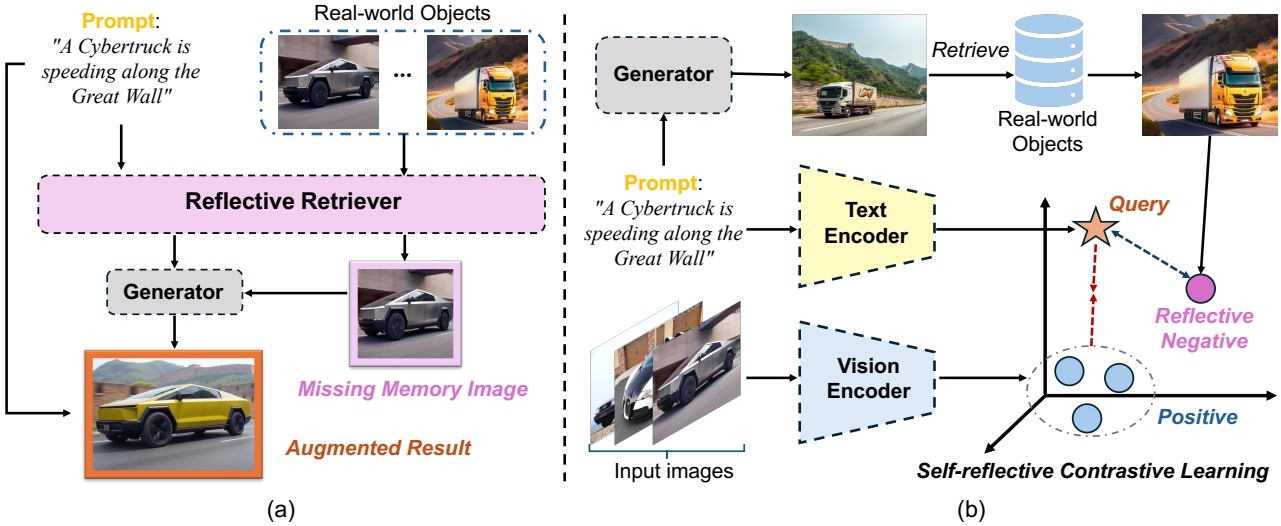

*Figure 2.* An overview of our RealRAG. **(a)** The pipeline of the real-object-based RAG. We propose the first real-object-based retrieval-augmented generation framework, which leverages real-world images to compensate for the knowledge gap inherent in generative models and augment realistic image generation, **(b)** The framework of self-reflective contrastive learning, which injects the generator's knowledge into the self-reflective negatives ensuring that the retrieved images compensate for the model's missing knowledge.

database is not direct and controllable for realistic image generation (Blattmann et al., 2022; Zheng et al., 2025). In this paper, we conduct a vision-based, real-object-based database, which is collected by realistic images from public real-world datasets, including ImageNet (Deng et al., 2009), Stanford Cars (Krause et al., 2013), Stanford Dogs (Dataset, 2011), and Oxford Flowers (Nilsback & Zisserman, 2008). In this way, we augment the realism of the generative images with the real-object-based database.

### 2.3. Contrastive Learning for Retrieval

Contrastive learning has emerged as a powerful method for retrieval tasks, leveraging the principle of learning representations by contrasting positive and negative samples (Khosla et al., 2020; Le-Khac et al., 2020). The approach aims to map semantically similar data points closer in the embedding space while simultaneously pushing dissimilar data points apart. Methods such as SimCLR (Chen et al., 2020) and MoCo (He et al., 2020) have popularized this framework in vision tasks, whereas in the domain of multi-modal retrieval, models including CLIP (Radford et al., 2021) have demonstrated their effectiveness by aligning textual and visual representations. Recent research has extended contrastive learning to various modalities, such as audio (Radford et al., 2021; Sun et al., 2023; Likhosherstov et al., 2021; Guzhov et al., 2022; Mahmud & Marculescu, 2023; Girdhar et al., 2023), video (Huang et al., 2023; Fang et al., 2021; Luo et al., 2022; Xue et al., 2022; Zhu et al., 2023), point cloud (Zhang et al., 2022; Zhu et al., 2022b; Huang et al., 2022; Guo et al., 2023), and tactile data (Yang et al., 2024;

Lei et al., 2024), thereby enhancing cross-modal retrieval capabilities. These methods often incorporate techniques (e.g., hard negative mining (Kalantidis et al., 2020; Robinson et al., 2020) and balanced learning (Zhu et al., 2022a; Liu et al., 2022)) to improve the quality of the learned embedding space. Despite its success in query-document matching, images that best match a text prompt may not be the most valuable references for text-to-image generative models (Zhang et al., 2021). Consequently, we propose a self-reflective contrastive learning approach, which retrieves images containing the missing knowledge of the generative models rather than selecting the most relevant images.

## 3. Methodology

**Problem Setting:** Given a textual prompt $T$ drawn from a text space $\mathcal{T}$, the text-to-image generation task aims to produce a corresponding image $I \in \mathcal{I}$ that accurately reflects the semantics of $T$. Formally, we model this task as learning a conditional distribution $p(I \mid T)$, which describes how likely an image $I$ is given the text $T$. In practice, we often parameterize this distribution with a neural generator $G_\theta$ (with parameters $\theta$), yielding:

$$P_\theta(I \mid T) \approx P(I \mid T), \qquad (1)$$

given a new text prompt $T$, the learned generator $G_\theta$ generate an image:

$$\hat{I} \sim P_\theta(I \mid T), \qquad (2)$$

Alternatively, one might produce a deterministic output by taking, for example, the mode of $P_\theta(I \mid T)$. In either case,

the goal is to ensure that $\hat{I}$ visually manifests the semantics conveyed by $T$.

**Overview:** An overview of RealRAG is shown in Fig. 2. Specifically, as shown in Fig. 2 (b), we first train the reflective retriever via self-reflective contrastive learning (**Sec. 3.1**). Powered by self-reflective contrastive learning, the reflective retriever retrieves images with the generator's missing knowledge. Subsequently, as shown in Fig. 2 (a), we utilize the recalled image as the reference for the realistic image generation, which effectively addresses the text-to-image generation's hallucinations (**Sec. 3.2**). We now describe these technical components in detail.

### 3.1. Self-reflective Contrastive Learning

We propose self-reflective contrastive learning to retrieve useful images for the generator. **Our key insight** is to train a retriever that *retrieves images staying off the generation space of the generator, yet closing to the representation of text prompts.* To this end, as shown in Fig. 2 (b), we first generate images from the given text prompts and then utilize the generated images as queries to retrieve the most relevant images in the real-object-based database. These most relevant images are utilized as reflective negatives. Concretely, we denote the generator as $G$, given a text prompt $T$, we generate image $I_{gen}$ and extract its feature embedding $Z_{gen}$ is by:

$$I_{gen} = G(T), \quad Z_{gen} = F_i(I_{gen}), \quad (3)$$

where $F_i$ is the vision encoder for extracting visual embedding from input images. We then select the self-reflective negative by ranking the images from the real-object-based database $\mathcal{I}$ with the cosine-similarity. We choose the highest one:

$$I_{neg} = \underset{I_{obj}^k \in \mathcal{I}}{\arg\max} \ \text{sim}\Big(F_i(I_{obj}^k), Z_{gen}\Big). \quad (4)$$

The reflective negative, with the highest cosine-similarity score, includes the original knowledge of the generator and empowers the retriever to capture the real-object-based images with missing knowledge of the generator. For training the retriever via self-reflective contrastive learning, we first extract object feature embeddings $\{Z_{obj}^1, Z_{obj}^2, ..., Z_{obj}^n\}$ for the real-object-based images $\{I_{obj}^1, I_{obj}^2, ..., I_{obj}^n\} \in \mathcal{I}$:

$$\{Z_{obj}^1, Z_{obj}^2, ..., Z_{obj}^n\} = F_i(\{I_{obj}^1, I_{obj}^2, ..., I_{obj}^n\}), \quad (5)$$

we select the embedding $Z_{obj}^{pos}$ of ground truth image, which is matched with the input text prompt $T$, from the object feature embeddings $\{Z_{obj}^1, Z_{obj}^2, ..., Z_{obj}^n\}$.

Subsequently, we encode text prompt $T$ as the query embedding $Z_q$ and the reflective negative $I_{neg}$ as the negative feature embedding $Z_{neg}$:

$$Z_q = F_t(T), \quad Z_{neg} = F_i(I_{neg}), \quad (6)$$

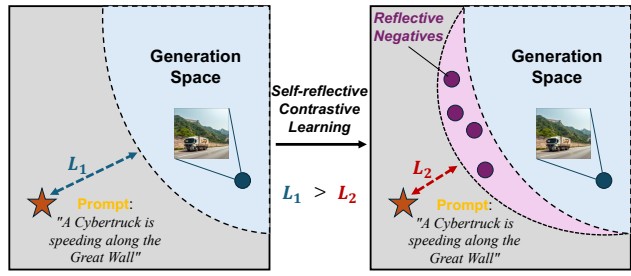

*Figure 3.* The comparison of generation space. We show the generation space of the normal RAG (**left**) and our RealRAG (**right**).

where $F_i$ is the vision encoder and $F_t$ is the text encoder (As shown in Fig. 2 (b)).

We perform our self-reflective contrastive learning by using normal in-batch negative and the reflective negative at the same time. We calculate the similarity between the text prompt and negatives:

$$\mathcal{D}_{neg}^{nor} = \sum_{j \neq pos}^N (exp(Z_q^T \cdot Z_{obj}^j/\tau)), \quad \mathcal{D}_{neg}^{ref} = exp(Z_q^T \cdot Z_{neg}), \quad (7)$$

where $N$ is the training batch size, $\mathcal{D}_{neg}^{nor}$ is the similarity between the text prompt and the in-batch negative, $\mathcal{D}_{neg}^{ref}$ is the similarity between the text prompt and the reflective negative.

Lastly, the overall loss of self-reflective contrastive learning is:

$$\mathcal{L} = -log \frac{exp(Z_q^T \cdot Z_{obj}^{pos}/\tau)}{exp(Z_q^T \cdot Z_{obj}^{pos}/\tau) + \mathcal{D}_{neg}^{nor} + \mathcal{D}_{neg}^{ref}}, \quad (8)$$

where $\tau$ is a temperature hyperparameter.

Powered by self-reflective contrastive learning, the reflective retriever integrates the missing knowledge of the generator, augmenting the realism and accuracy of the generated images and effectively addressing the hallucination problems in the text-to-image generation.

### 3.2. Real-object-based Retrieval-augmented Generation

Expanding upon our trained reflective retriever, we subsequently retrieve the augmented real-object-based images $I_{ref}$ based on the given text prompt $T$ for the generator:

$$I_{ref} = Retriever(T, \{I_{obj}^1, I_{obj}^2, ...I_{obj}^n\}), \quad (9)$$

where $Retriever$ is the trained reflective retriever, and $n$ in the scale of real-object-base database.

We then generate image $I_{res}$ based on the text prompt $T$ and the augmented image $I_{ref}$:

$$I_{res} = G(T, I_{ref}), \quad (10)$$

where $G$ is the generator. Our real-object-based RAG improves the knowledge of the generator and significantly

extends the generation space for the frozen generator. As shown in Fig. 3, powered by Eq. 8, the reference image is distributed in the distribution space outside the generator's generation space and close to the text prompt embedding. In this way, the generation space of our real-object-based RAG can effectively expand toward the text prompt embedding.

### 3.3. Implementation

Our RealRAG can be flexibly implemented with different existing generators, including U-Net-based diffusion models, DiT-based diffusion models, and auto-regressive models. It is the first work to build *a unified RAG framework for all types of text-to-image generative models*.

**Real-object-based Database.** We collect our real-object-based database from wide-use real-world datasets, including ImageNet (Deng et al., 2009), Stanford Cars (Krause et al., 2013), Stanford Dogs (Dataset, 2011), and Oxford Flowers (Nilsback & Zisserman, 2008). We use the training set of these datasets to conduct our database.

**Generator.** We use existing text-to-image generative models as the generator to implement our RealRAG. Concretely, we implement our RealRAG with the following models: U-Net-based diffusion models (SD V2.1 (Rombach et al., 2022a), SD XL (Podell et al., 2023)), DiT-based diffusion models (SD V3 (Esser et al., 2024), Flux (BlackForest, 2024)), and auto-regressive models (OmniGen (Xiao et al., 2024), Emu (Sun et al., 2024)). Specifically, for autoregressive models, they can directly generate images from the image condition. For diffusion models, we utilize ControlNet, which introduces a branch to stable diffusion, enabling the inclusion of additional inputs, to input image-based conditions. For example. First, we retrieve and sort the closest images. Next, we input the selected images into the ControlNet branch to control specific elements during the image synthesis process.

**Training Details.** We train our reflective retriever based on the pre-trained CLIP model (Radford et al., 2021). We add a simple MLP layer at the end of the vision encoder of the CLIP model, to map the visual embeddings outside the generation space of the generator and close to the input prompt embedding. We utilize the frozen text encoder from the CLIP model as our text encoder.

## 4. Experiment

### 4.1. Datasets and Implementation Details

**Datasets and Benchmarks.** We evaluate our proposed RealRAG on three fine-grained real-world image datasets, including Stanford Cars (Krause et al., 2013), Stanford Dogs (Dataset, 2011), and Oxford Flowers (Nilsback & Zisserman, 2008). We use the test sets of these datasets

to validate the realism of the generated images (Sec. 4.2). Furthermore, to validate the ability of RealRAG to generate unseen novel objects, we also test our model on recently introduced novel objects (Sec.4.3).

**Evaluation Metrics.** We employ FID, CLIP-T, and CLIP-I to compare the visual quality and realism of generated images from different methods. Specifically, FID measures how closely the distribution of generated images matches that of real images by comparing their feature statistics in the latent space of Inception V3. CLIP-T uses CLIP model to assess how well the generated image matches with its text prompt, essentially quantifying text-image correspondence. CLIP-I, on the other hand, focuses on measuring image-image similarity through CLIP, often comparing a generated image to a reference or target image, thereby evaluating visual fidelity or consistency across images.

### 4.2. Fine-grained Object Generation

#### 4.2.1. EXPERIMENT SETUP

To evaluate the realism of the generated images, we use the text prompt ["A photo of a [CLASS NAME]"] for image generation. We generate ten images for each class to get more reliable results by multiple sampling. We use fine-grained real-world datasets to evaluate the realism of the generated images by calculating the FID score, CLIP-T score, and CLIP-I score between the ground truth images and the generated images.

#### 4.2.2. GENERATIVE RESULT

**Quantitative Results.** In Tab. 1, we apply our method with all the types of SoTA text-to-image generators. Our RealRAG demonstrates significant performance gains with all these models on the three benchmarks, including an average **gain of *6.19%*** on the Stanford Cars, **a *3.62%* gain** on the Stanford Dogs, and **an *8.94%* gain** on the Oxford Flowers. As shown in Tab. 1, our RealRAG achieves the **most significant improvement** (a *9.90%* gain on average) with the auto-regressive model, which shows the potential of our RealRAG to enhance the development of the large-scale auto-regressive model. To further evaluate the generative quality and realism of fine-grained objects, we utilize a pre-trained classification model (OpenCLIP (Cherti et al., 2022)) to calculate the classification accuracy for the generated images. As shown in Tab. 2, our RealRAG achieves considerable improvements of the classification performance, *e.g.*, **a *3.89%* gain** for the auto-regressive model.

**Qualitative Results.** We show the qualitative results in Fig. 4. The visual results show the ability of our RealRAG to *overcome hallucinations and significantly improve the realism and quality of fine-grained realistic image generation*. Specifically, for the autoregressive model, In the

*Table 1.* Evaluation of fine-grained object generation. We report the quantitative results on the Stanford Cars (Krause et al., 2013), Stanford Dogs (Dataset, 2011), and Oxford Flowers (Nilsback & Zisserman, 2008) benchmarks.

| Type | Method | Stanford Cars | | | Stanford Dogs | | | Oxford Flowers | | |
|---|---|---|---|---|---|---|---|---|---|---|
| | | CLIP-I ↑ | CLIP-T ↑ | FID ↓ | CLIP-I ↑ | CLIP-T ↑ | FID ↓ | CLIP-I ↑ | CLIP-T ↑ | FID ↓ |
| AR Model | OmniGen (Xiao et al., 2024) | 55.69 | 11.25 | 77.35 | 72.47 | 17.92 | 67.94 | 82.78 | 20.72 | 66.22 |
| | OmniGen W. Ours | 57.11 | 14.48 | 74.98 | 78.74 | 18.30 | 63.10 | 84.65 | 20.98 | 54.42 |
| | Emu (Sun et al., 2024) | 61.41 | 13.49 | 86.73 | 80.51 | 19.52 | 48.75 | 81.88 | 21.17 | 84.33 |
| | Emu W. Ours | 63.07 | 15.30 | 70.55 | 80.39 | 19.60 | 45.04 | 85.52 | 22.42 | 63.85 |
| | Δ | +1.54 | +2.52 | -9.28 | +3.08 | +0.23 | -4.28 | +2.76 | +0.75 | -16.14 |
| U-Net-based Diffusion Model | SD V2.1(Rombach et al., 2022a) | 61.48 | 14.16 | 64.99 | 68.80 | 17.69 | 49.60 | 82.97 | 20.76 | 69.17 |
| | SD V2.1 W. Ours | 64.50 | 15.84 | 58.98 | 79.06 | 18.57 | 43.93 | 88.43 | 21.77 | 60.76 |
| | SDXL (Podell et al., 2023) | 59.53 | 13.03 | 61.98 | 77.10 | 18.05 | 49.28 | 81.08 | 19.59 | 55.20 |
| | SDXL W. Ours | 64.71 | 15.16 | 60.95 | 77.79 | 18.31 | 45.31 | 85.63 | 20.92 | 47.41 |
| | Δ | +4.10 | +1.91 | -3.52 | +5.48 | +0.57 | -4.82 | +5.01 | +1.17 | -8.10 |
| DiT-based Diffusion Model | SD V3 (Esser et al., 2024) | 59.43 | 12.61 | 59.94 | 80.28 | 18.56 | 53.37 | 83.23 | 20.19 | 63.46 |
| | SD V3 W. Ours | 60.01 | 14.80 | 54.60 | 81.04 | 18.62 | 53.28 | 82.68 | 20.73 | 60.89 |
| | Fux (BlackForest, 2024) | 60.50 | 13.70 | 58.47 | 80.16 | 18.36 | 45.71 | 82.18 | 20.19 | 55.93 |
| | Flux W. Ours | 62.81 | 14.46 | 52.28 | 83.53 | 18.64 | 42.25 | 86.18 | 21.46 | 53.32 |
| | Δ | +1.45 | +1.48 | -5.77 | +2.07 | +0.17 | -1.78 | +1.73 | +0.91 | -2.59 |

*Figure 4.* The visual results of fine-grained object generation. We visually compare the images generated by the original generators and our RealRAG. We also add real-world images for reference.

prompt ["Audi TT Hatchback 2011"], the Real-RAG model produces a highly realistic rendering of the vehicle, capturing intricate details like the side mirrors and headlights, which are closer to the real object compared to the original AR model with significant hallucination in the car's side. For the U-Net-based diffusion model, In prompt ["Silverbush"], RealRAG ensures the correct flower structure and vibrant color tones, whereas the original model struggles with maintaining fine-grained details like petal shape. Lastly, compared with the original generator, RealRAG delivers a compelling representation of

the breed's signature coat patterns and body posture from the ["Bluetick"] prompt. The results demonstrate that RealRAG generates high realism and correct objects and show the flexible application of the SoTA generators.

### 4.3. Unseen Novel Object Generation

#### 4.3.1. EXPERIMENT SETUP

To evaluate the ability to generate unseen novel objects, we collect several recently introduced objects (e.g., the Cybertruck) to form the input prompts. Specifically, we use a

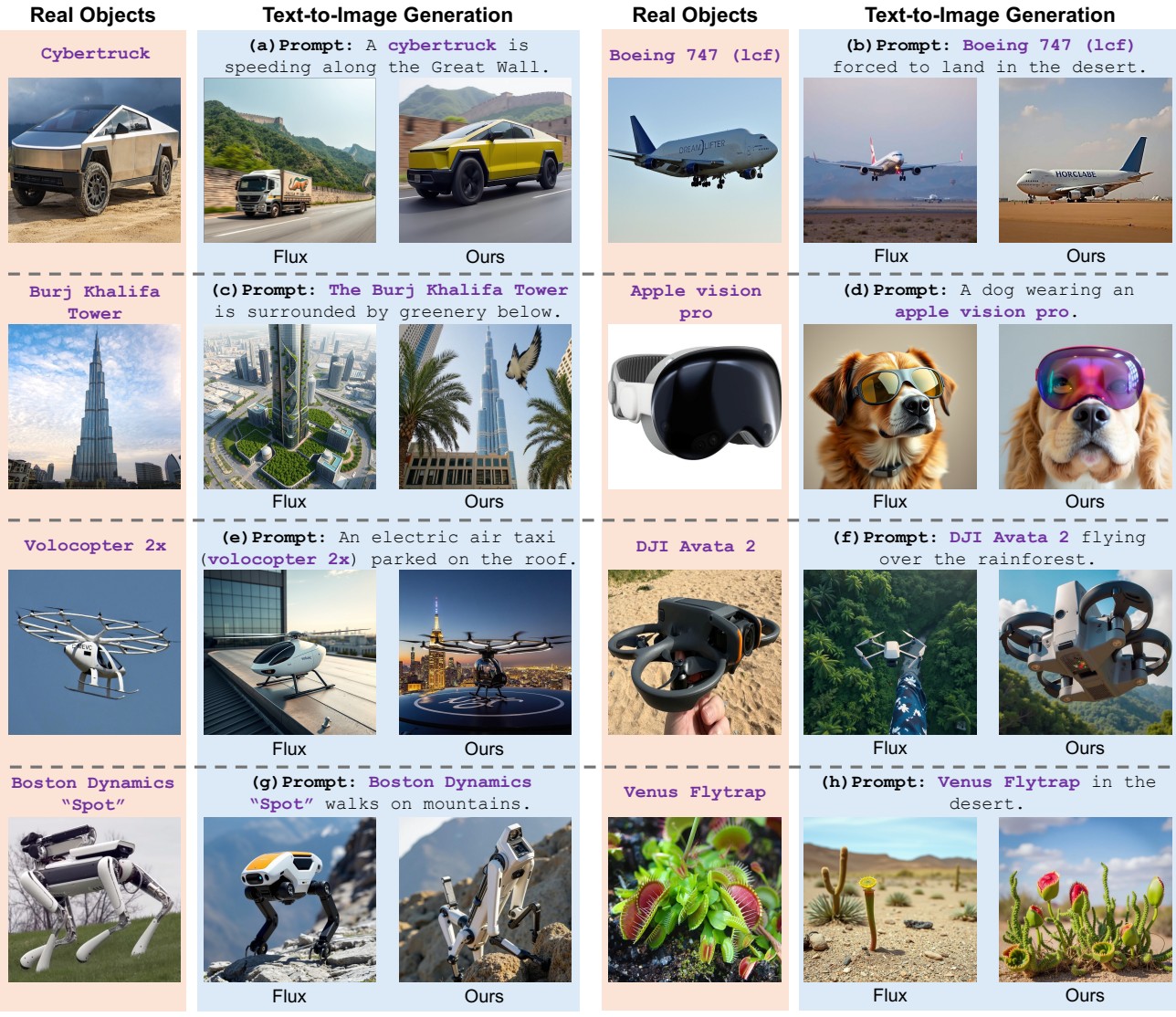

*Figure 5.* The visual results of unseen novel object generation.

LLM (e.g., ChatGPT) to generate prompts based on these novel objects. For the image database, we use the images from the Internet (Google Images), which include the novel objects. To ensure fairness in our comparison, we conduct both qualitative evaluations and human evaluations.

### 4.3.2. GENERATIVE RESULT

**Qualitative Results.** In Fig. 5, we present the visual comparison of the SoTA model Flux (BlackForest, 2024) and Our RealRAG. The visual results demonstrate the significant improvement in the unseen object generation of our RealRAG. As the case `["Cybertruck"]` shows, compared with the original result generated by Flux, our RealRAG generates the realistic shape for the Cybertruck and also synthesis the sense of `["speeding along the Great Wall"]`; As the case `["Boston Dynamics Spot"]`,

although the Flux can generate the sense of `["a robot walks on mountains"]`, the shape and type of the `["Boston Dynamics Spot"]` is inaccurate. Our RealRAG can generate the correct and realistic objects, which shows the significant improvement of our RealRAG for unseen noval object generation.

**Human Evaluation.** We conducted a human evaluation to assess the accuracy of the generated images. Data acquisition primarily revolved around participants' subjective assessments of the generative accuracy of the unseen novel objects. We involved **26 participants** in our evaluation (*More details are included in the Appendix.*). We use a **7-point Likert scale** to evaluate the accuracy of the generated images in the human's view. As shown in Fig. 6, we ask participants to give a score from 1 (low accuracy) to 7 (high accuracy) for the 4 sets of images. The results show that

*Table 2.* The classification results of fine-grained object generation. We use the pre-trained classification model (OpenCLIP (Cherti et al., 2022)) to test the classification accuracy of the generated images from the three datasets. A higher accuracy indicates the generated images are more similar to real images. We report the average accuracy score (*the full results of the three datasets are shown in Tab. 5 of the Appendix*).

| Type | Method | Average Acc. ↑ |
|---|---|---|
| Autoregressive Model | OmniGen (Xiao et al., 2024) | 35.26 |
| | OmniGen W. Ours | 38.96 |
| | Emu (Sun et al., 2024) | 35.81 |
| | Emu W. Ours | 39.90 |
| | Δ | +3.89 |
| U-Net-based Diffusion Model | SD V2.1(Rombach et al., 2022a) | 37.45 |
| | SD V2.1 W. Ours | 40.23 |
| | SDXL (Podell et al., 2023) | 36.32 |
| | SDXL W. Ours | 40.49 |
| | Δ | +3.48 |
| DiT-based Diffusion Model | SD V3 (Esser et al., 2024) | 36.09 |
| | SD V3 W. Ours | 39.03 |
| | Fux (BlackForest, 2024) | 37.29 |
| | Flux W. Ours | 40.37 |
| | Δ | +2.99 |

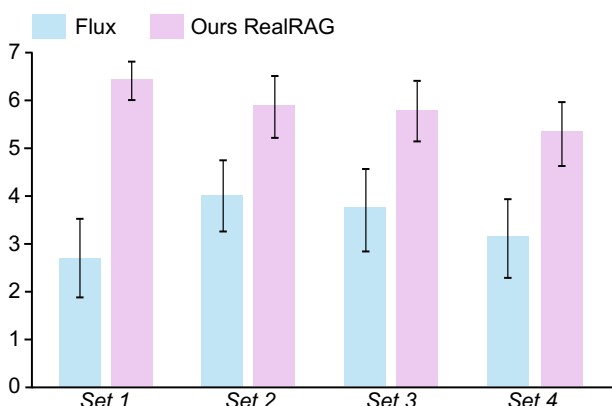

*Figure 6.* The 7-point Likert scale of the human evaluation. The participant is asked to score the generated images from 1 (low accuracy) to 7 (high accuracy), according to given text prompts.

all participants consistently rated the quality of RealRAG-generated images with a mean score exceeding 5, indicating significant performance gains compared to Flux. RealRAG exhibited a relatively small variance across samples, highlighting its proficiency in unseen novel object generation. Overall, *the combination of higher mean scores and lower variance demonstrates that RealRAG is not only better at generating higher-realism images but also more reliable, as participants consistently rated its outputs highly.*

## 5. Ablation Study

To investigate and analyze the effectiveness of our proposed RealRAG, we first perform an ablation study on the

*Table 3.* Evaluation of fine-grained object generation. We report the quantitative results on the Stanford Cars (Krause et al., 2013).

| Generator | Method | CLIP-I ↑ | CLIP-T ↑ | FID ↓ |
|---|---|---|---|---|
| Emu (Sun et al., 2024) | Zero-RAG | 61.84 | 14.06 | 80.75 |
| | Normal-RAG | 62.20 | 14.31 | 76.85 |
| | RealRAG | **63.07** | **15.30** | **70.55** |
| SD V2.1 (Rombach et al., 2022a) | Zero-RAG | 62.59 | 14.87 | 61.67 |
| | Normal-RAG | 63.05 | 15.03 | 60.77 |
| | RealRAG | **64.50** | **15.84** | **58.98** |
| Flux (BlackForest, 2024) | Zero-RAG | 61.20 | 13.96 | 54.25 |
| | Normal-RAG | 61.72 | **14.52** | 54.04 |
| | RealRAG | **62.81** | 14.46 | **52.28** |

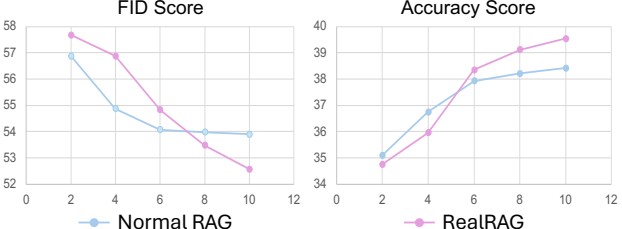

*Figure 7.* The results present the performance comparison of Normal RAG and our RealRAG with the checkpoint from the second, fourth, sixth, eighth, and tenth epoch.

fine-grained object generation setting. As shown in Table 3, we compare the generative performance of three variants: Zero-shot RAG, Normal RAG, and RealRAG. Specifically, Zero-shot RAG is a baseline approach that employs a pre-trained CLIP model to retrieve relevant images based on cosine similarity, whereas Normal RAG trains the retriever via contrastive learning to retrieve the most relevant images. We evaluate each approach using three different generators—Emu (Sun et al., 2024) (AR model), SDXL (Podell et al., 2023) (U-Net-based diffusion model), and Flux (BlackForest, 2024) (DiT-based diffusion model). The results highlight the effectiveness of our proposed self-reflective contrastive learning. Compared with the Normal RAG framework, RealRAG achieves substantial improvements in both generation realism and accuracy. In addition, we employ a classification model to measure the classification accuracy of the generated images (*see Table 5 in the Appendix*), which further demonstrates the robust performance of RealRAG for fine-grained, realistic image generation.

To further investigate the effectiveness of the reflective negative, we evaluate the performance of the checkpoints from the second, fourth, sixth, eighth, and tenth epochs. As shown in Fig. 7, while our RealRAG does not improve as rapidly as retrieval frameworks relying solely on similarity in the early stages of training, it ultimately surpasses the generative bottleneck in later training stages, powered by the reflective negative.

Lastly, the results of the t-SNE visualization in Fig. 8 reveal the differences between the representation spaces con-

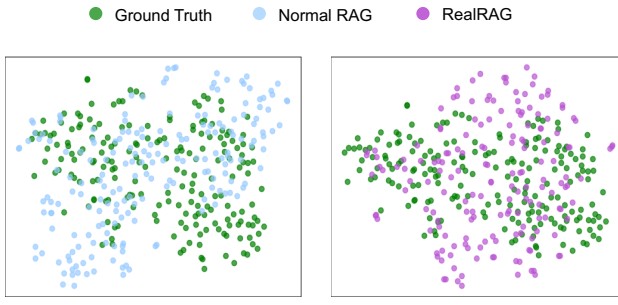

● Ground Truth  ● Normal RAG  ● RealRAG

*Figure 8.* The t-SNE visualization of the generated images.

structed by normal RAG and our RealRAG. The visual results demonstrate that the generative space of RealRAG expands more in the direction of ground-truth images.

## 6. Discussion

**Research Purpose:** Different from RDMs (Rombach et al., 2022b; Blattmann et al., 2022; Sheynin et al., 2022; Chen et al., 2022) that use retrieval-augmented techniques to train or fine-tune a diffusion model and achieve out-of-distribution (OOD) image generation by switching databases, our RealRAG aims to use retrieval-augmented techniques to train or fine-tune a diffusion model and achieve out-of-distribution (OOD) image generation by switching databases.

**Unseen novel objects:** These refer to objects that appear after the generative models and retrieval models are trained. The generative model cannot generate these objects, and the retrieval model can't easily retrieve relevant references by similarity. This is a much more challenging issue.

**Hallucination When Generating Fine-Grained Objects:** Existing SoTA t2i models are pre-trained on large-scale text-image paired datasets. As a result, they tend to produce hallucinations, such as inaccuracy or unrealistic features, when generating fine-grained objects. This is a problem inherent to large generative models.

## 7. Conclusion

In this paper, we proposed RealRAG, the **first real-object-based retrieval-augmented generation framework**. Our RealRAG augmented fine-grained and unseen novel object generation by learning and retrieving real-world images to overcome the knowledge gaps of generative models. Our RealRAG achieved **remarkable performance boosts** and is compatible with **all types of generative models**. Additionally, we examined the potential of the RAG framework for text-to-image generation. For future work, we will continue to explore more efficient RAG frameworks for the text-to-image generation task and other generation tasks.

## Impact Statement

This paper advances the field of machine learning, with a particular focus on text-to-image generation. It addresses the prevalent challenge of hallucinations and distortions in generated images, which arise from the limited knowledge stored in the fixed parameters of generative models—an issue frequently encountered in real-world applications. While there are potential societal implications, we feel none need to be specifically highlighted here.

## Acknowledgments

This work was supported by Open Project Program of Guangxi Key Laboratory of Digital Infrastructure (Grant Number: GXDIOP2024015); Guangdong Provincial Department of Education Project (Grant No.2024KQNCX028); Scientific Research Projects for the Higher-educational Institutions (Grant No.2024312096), Education Bureau of Guangzhou Municipality; Guangzhou-HKUST(GZ) Joint Funding Program (Grant No.2025A03J3957), Education Bureau of Guangzhou Municipality.

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

# A. Appendix

## A.1. More Experimental Results

We show the full classification results of fine-grained object generation in Tab. 4. The results demonstrate that RealRAG effectively enhances the generative quality and realism. Notably, the augment of our RealRAG is not limited to the original generative ability of the generators. For example, although the SoTA model, Flux (BlackForest, 2024) achieves the best performance on the Stanford Car, *e.g.*35.78 Accuracy score, our RealRAG also effectively enhances the generative quality and realism (a ***4.20%*** gain).

*Table 4.* The full classification results of fine-grained object generation.

| Type | Method | Stanford Cars | Stanford Dogs | Oxford Flowers | Average Acc. ↑ |
|---|---|---|---|---|---|
| Autoregressive Model | OmniGen (Xiao et al., 2024) | 32.80 | 40.93 | 32.04 | 35.26 |
| | OmniGen W. Ours | 37.45 | 44.71 | 34.72 | 38.96 |
| | Emu (Sun et al., 2024) | 33.94 | 41.32 | 32.18 | 35.81 |
| | Emu W. Ours | 40.24 | 44.33 | 35.13 | 39.90 |
| | Δ | +5.48 | +3.40 | +2.82 | +3.89 |
| U-Net-based Diffusion Model | SD V2.1(Rombach et al., 2022a) | 35.21 | 42.35 | 34.79 | 37.45 |
| | SD V2.1 W. Ours | 39.43 | 46.10 | 35.15 | 40.23 |
| | SDXL (Podell et al., 2023) | 35.20 | 41.06 | 32.69 | 36.32 |
| | SDXL W. Ours | 40.55 | 44.76 | 36.17 | 40.49 |
| | Δ | +4.79 | +3.73 | +1.92 | +3.48 |
| DiT-based Diffusion Model | SD V3 (Esser et al., 2024) | 33.90 | 41.26 | 33.12 | 36.09 |
| | SD V3 W. Ours | 38.72 | 44.24 | 34.05 | 39.03 |
| | Fux (BlackForest, 2024) | 35.78 | 42.11 | 33.98 | 37.29 |
| | Flux W. Ours | 39.98 | 45.84 | 35.29 | 40.37 |
| | Δ | +4.51 | +3.36 | +1.12 | +2.99 |

## A.2. More Results in Ablation Study

We show the ablation study of fine-grained classification in Tab. 5. Specifically, Zero-shot RAG is the straightforward pipeline of the RAG, which simply uses the pre-trained CLIP model to retrieve relevant images by cosine-similarity; Normal RAG trains the retriever via contrastive learning to retrieve the most relevant images. The results show the effectiveness of our proposed self-reflective contrastive learning. Compared with the normal RAG framework, our RealRAG achieves significant improvements in generation realism and accuracy.

*Table 5.* The ablation study of fine-grained classification on the three datasets.

| Generator | Method | Stanford Cars | Stanford Dogs | Oxford Flowers | Average Acc. ↑ |
|---|---|---|---|---|---|
| Emu (Sun et al., 2024) | Zero-RAG | 37.18 | 42.88 | 34.27 | 38.11 |
| | Normal-RAG | 38.75 | 43.09 | 35.10 | 38.98 |
| | RealRAG | **40.24** | **44.33** | **35.13** | **39.90** |
| SD V2.1 (Rombach et al., 2022a) | Zero-RAG | 38.64 | 44.07 | 33.70 | 38.80 |
| | Normal-RAG | 39.07 | 44.92 | **35.21** | 39.73 |
| | RealRAG | **39.43** | **46.10** | 35.15 | **40.23** |
| Flux (BlackForest, 2024) | Zero-RAG | 36.27 | 44.31 | 34.59 | 38.39 |
| | Normal-RAG | 37.84 | 45.01 | 34.96 | 39.27 |
| | RealRAG | **39.98** | **45.84** | **35.29** | **40.37** |

## A.3. Details and More Results in Human Evaluation

We conducted a human evaluation to assess the realism and accuracy of the generated images. Data acquisition primarily revolved around participants' subjective assessments of the realism in fine-grained image generation and accuracy in unseen novel image generation. The evaluation consists of two tasks: **(1)** Text-to-image matching task and **(2)** Real/False task. The

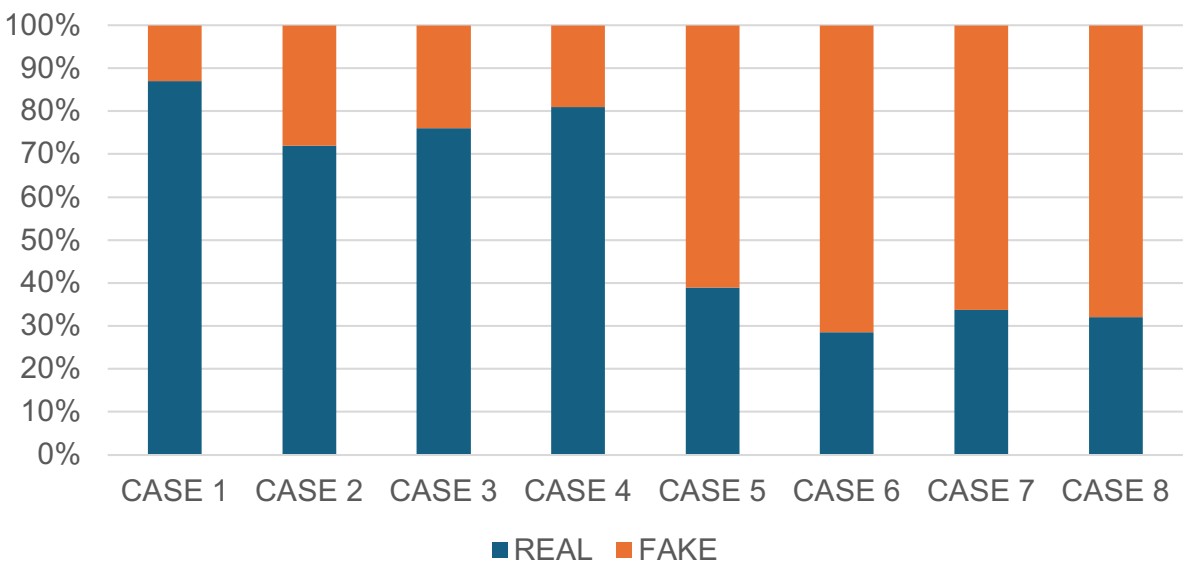

*Figure 9.* The results of the Real/fake task. Case 1,2,3,4 are generated by our RealRAG and case 5,6,7,8 are generated by the original generators.

human evaluation procedures were approved by the ethical committee.

**Participants.** The evaluation involved 26 participants, of whom 69.23% were aged 18-24 and 30.77% were aged 25-34. The gender distribution was 53.85% male and 46.15% female. Also, 57.69% of the participants had previous experience with generative models, and 42.31% had no experience using AI generative models in the last six months.

**Tasks and Measurement.** We designed two tasks to evaluate the effectiveness of RealRAG for its ability to generate fine-grained and unseen novel objects. The two tasks include: **(1)** Text-to-image matching task and **(2)** Real/False task.

- **Text-to-image matching task.** In this task, we collected four sets of images, and for each set, there were two images, one was generated by the original Flux model (BlackForest, 2024), and the other was generated by our RealRAG. We then asked the participants to rate how well the text prompt and the generative image matched via a 7-point Likert scale. Images were presented in a random order, and the specific cases are shown in Fig. 10.

- **Real/Fake task.** In this task, we selected eight cases, of which, four of them are generated by original generators and others are generated by our RealRAG. Furthermore, we asked the participants whether the images were real or fake images (generated by AI). We show the task details in Fig. 11.

**Results.** We show the evaluation results of task **(1)** in Fig.6 of the main paper. For the task (2), we present the human-evaluation results in Fig.9. The results show that the images generated by our RealRAG have got higher scores from the participants. To summarize, more than 70% of the participants found our images to be realistic, which demonstrates the strong performance of our RealRAG for realistic image generation.

## B. More Visualization

### B.1. Qualitative Feature Visualization

We show more t-SNE visualization results in Fig. 12, which shows that the generative space of RealRAG expands more in the direction of ground-truth images.

### B.2. Generative Results

We show more generative results for fine-grained generative results in Fig. 13. The results show the generative ability of our RealRAG in realistic image generation.

**text: A cybertruck is speeding along the Great Wall.**

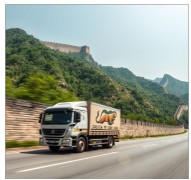

Please review the left image and text carefully.
How would you rate the matching of the images to the text? On a scale of 1 (low) to 7 (high).

**text: Boeing 747 (lcf) forced to land in the desert.**

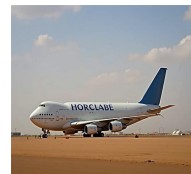

Please review the left image and text carefully.
How would you rate the matching of the images to the text? On a scale of 1 (low) to 7 (high).

**Set 1**

**text: The Burj Khalifa Tower is surrounded by greenery below.**

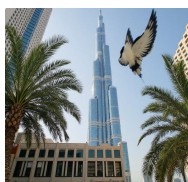

Please review the left image and text carefully.
How would you rate the matching of the images to the text? On a scale of 1 (low) to 7 (high).

**text: A dog wearing an apple vision pro.**

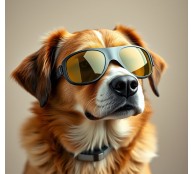

Please review the left image and text carefully.
How would you rate the matching of the images to the text? On a scale of 1 (low) to 7 (high).

**Set 2**

**text: An electric air taxi (volocopter 2x) parked on the roof.**

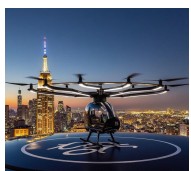

Please review the left image and text carefully.
How would you rate the matching of the images to the text? On a scale of 1 (low) to 7 (high).

**text: DJI Avata 2 flying over the rainforest.**

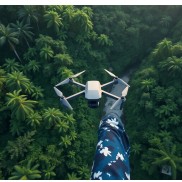

Please review the left image and text carefully.
How would you rate the matching of the images to the text? On a scale of 1 (low) to 7 (high).

**Set 3**

**text: Boston Dynamics "Spot" walks on mountains.**

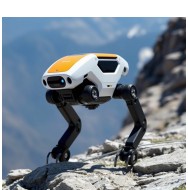

Please review the left image and text carefully.
How would you rate the matching of the images to the text? On a scale of 1 (low) to 7 (high).

**text: Venus Flytrap in the desert.**

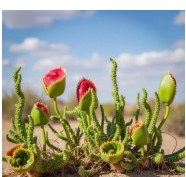

Please review the left image and text carefully.
How would you rate the matching of the images to the text? On a scale of 1 (low) to 7 (high).

**Set 4**

*Figure 10.* The cases in **Text-to-image matching task** of the human evaluation.

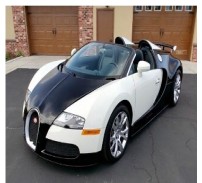

Please review the left image and text carefully.
How would you rate the realism of the images? On a scale of 1 (low) to 7 (high).

**Case 1**

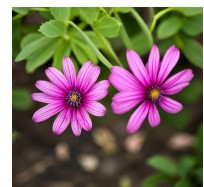

Please review the left image and text carefully.
How would you rate the realism of the images? On a scale of 1 (low) to 7 (high).

**Case 2**

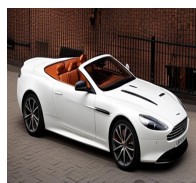

Please review the left image and text carefully.
How would you rate the realism of the images? On a scale of 1 (low) to 7 (high).

**Case 3**

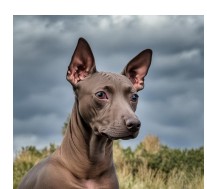

Please review the left image and text carefully.
How would you rate the realism of the images? On a scale of 1 (low) to 7 (high).

**Case 4**

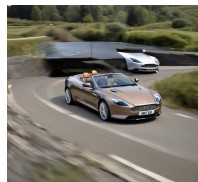

Please review the left image and text carefully.
How would you rate the realism of the images? On a scale of 1 (low) to 7 (high).

**Case 5**

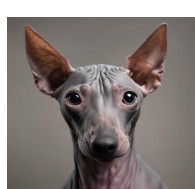

Please review the left image and text carefully.
How would you rate the realism of the images? On a scale of 1 (low) to 7 (high).

**Case 6**

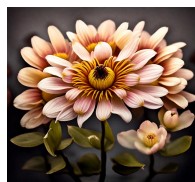

Please review the left image and text carefully.
How would you rate the realism of the images? On a scale of 1 (low) to 7 (high).

**Case 7**

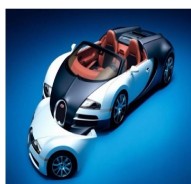

Please review the left image and text carefully.
How would you rate the realism of the images? On a scale of 1 (low) to 7 (high).

**Case 8**

*Figure 11.* The cases in **Real/Fake task** of the human evaluation. Case 1,2,3,4 are generated by our RealRAG and case 5,6,7,8 are generated by the original generators.

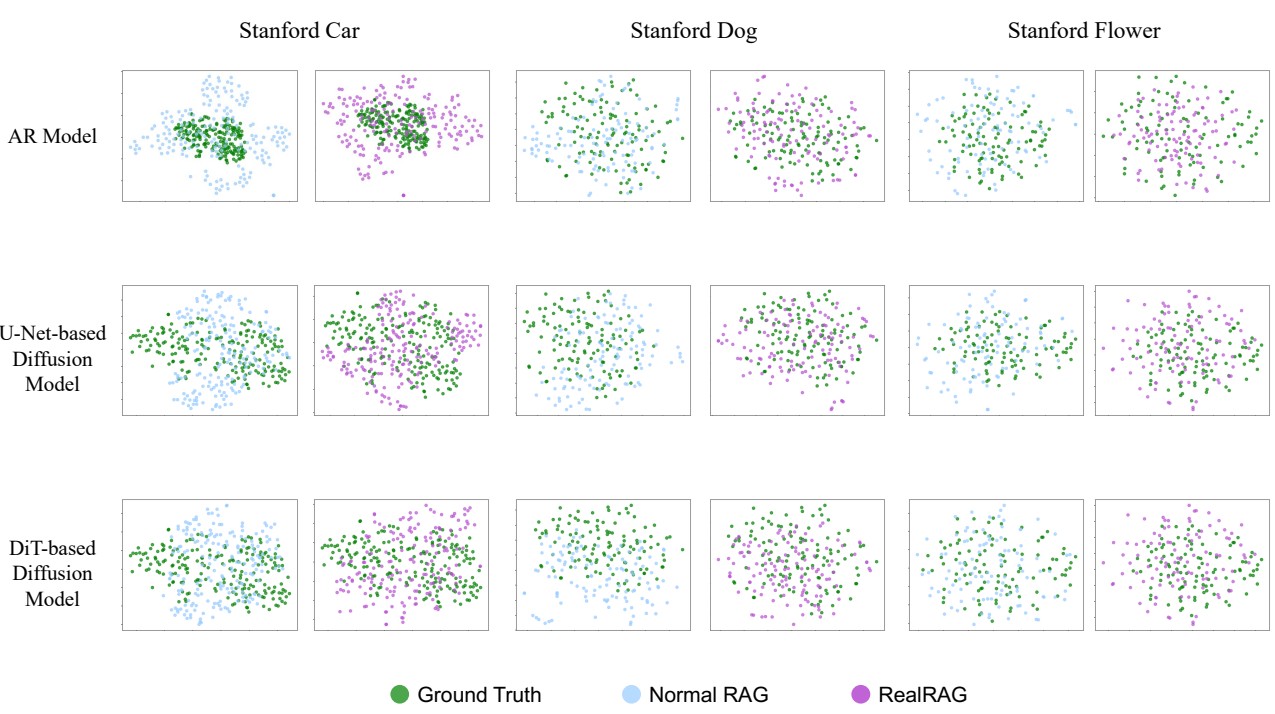

*Figure 12.* More t-SNE visualization results of generated images.

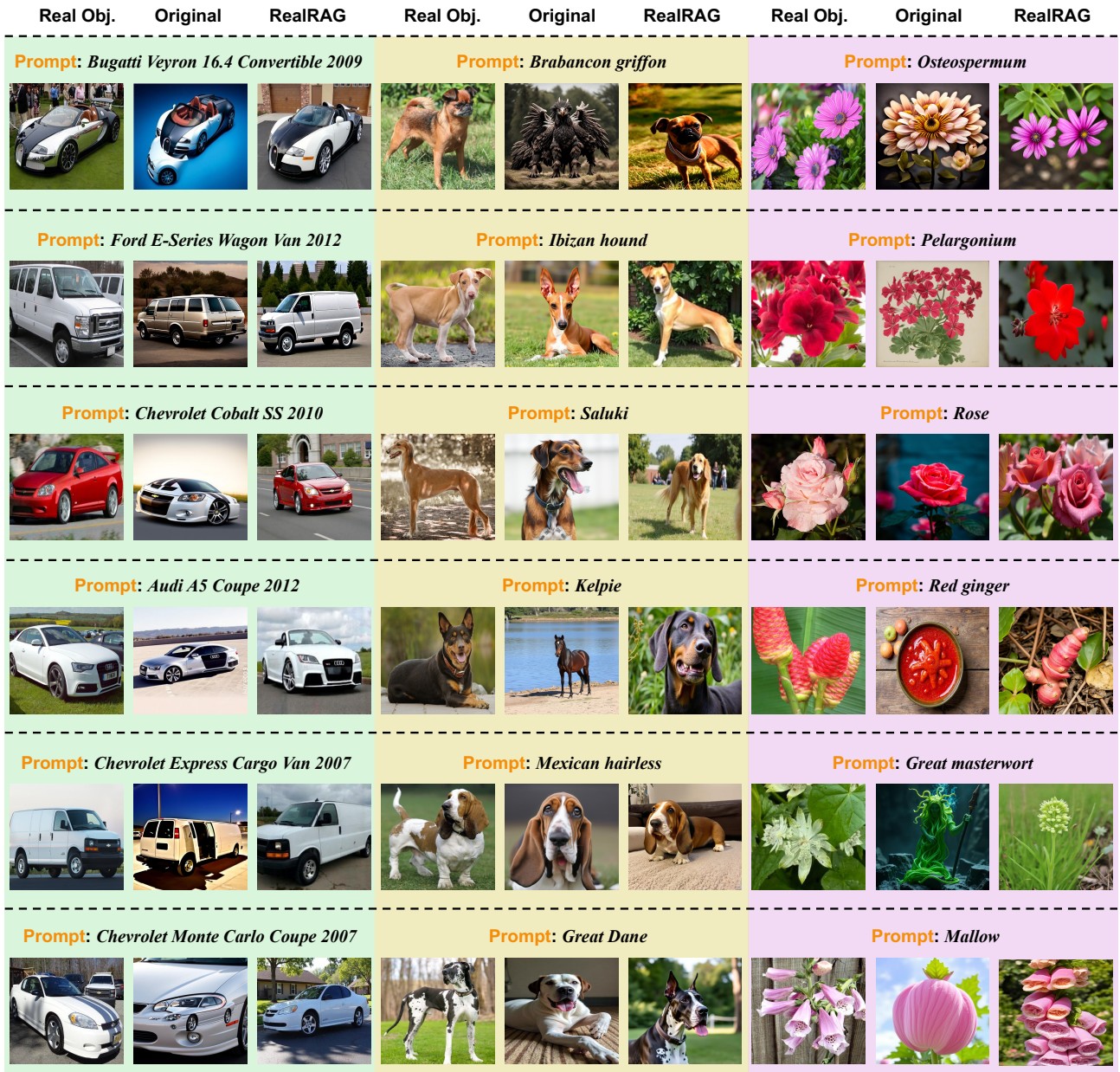

*Figure 13.* More visual results of fine-grained object generation.

## C. Application Value

For existing commercial text-to-image generative models, training them is cost-prohibitive. Therefore, a pipeline is needed to integrate real-time updated data from the internet into the generative model without any retraining, enabling the generation of unseen novel objects. On the other hand, in specific application scenarios such as advertising creation and multi-modal generation (Tang et al., 2023; Lyu et al., 2024a), users need generated images that meet their design requirements, while also ensuring that products (fine-grained objects) within the image remain realistic. This requires generative models to have the ability to generate both open-domain and fine-grained objects. Therefore, RealRAG focuses on reducing hallucinations in large t2i generators through RAG technology, enabling open-domain generators to generate specific fine-grained objects.

Beyond text-to-image (T2I) generation itself, several downstream tasks depend on and extend the same technology, like 3D generation (Li et al., 2024; Jiang et al., 2023a; Li et al., 2023) and video generation (Singer et al., 2022; Gu et al., 2025; Wang et al., 2025). For example, (i) text-to-3D synthesis (Poole et al., 2022; Jiang et al., 2024a;b; Bai et al., 2023) generally requires a strong T2I backbone, while (ii) image-to-3D reconstruction (Jiang et al., 2025; Hong et al., 2023; Tang et al., 2024) often begins with an image created by a T2I model.

