# OpenReview forum: "RealRAG: Retrieval-augmented Realistic Image Generation via Self-reflective Contrastive Learning"
_ICML.cc/2025/Conference — ICML 2025 poster_

### Official Review · Reviewer_HZx9 · 2025-03-14

**Overall Recommendation:** 3

**Summary:**

This paper introduces RealRAG, a retrieval-augmented generation (RAG) framework that enhances text-to-image models by retrieving real-world images to improve realism, accuracy, and faithfulness to fine-grained and unseen objects. Unlike conventional text-to-image models that suffer from hallucinations due to their fixed knowledge within model parameters, RealRAG retrieves real-object images and incorporates them into the generation process. The core innovation is the Reflective Retriever, trained via self-reflective contrastive learning, which aims to integrate missing memory rather than just the most similar images. The framework is designed to be modular and compatible with various generative models, including diffusion models (U-Net-based, DiT-based) and autoregressive models.

**Claims And Evidence:**

The claim of "the first real-object-based retrieval-augmented generation framework" is arguable. There are some previous works in real image-based RAG.

**Essential References Not Discussed:**

As mentioned above, [1] and [3] could be mentioned and discussed. Though [2] is mentioned, "the text database is not direct
and controllable for realistic image generation" seems not proper for describing [2]. A better description is desired.

**Experimental Designs Or Analyses:**

Yes.

**Methods And Evaluation Criteria:**

The experiments are limited to datasets that are mainly focused on one category, e.g., Stanford Cars, Stanford Dogs, and Oxford Flowers. The ability of the proposed model on general cases needs to be further studied.

**Other Comments Or Suggestions:**

What does the "!" mean in "SD V2.1"? (Table 1 and 2).

**Other Strengths And Weaknesses:**

1. The paper is well-written and easy to follow.

2. The qualitative results clearly demonstrate the advantages of the proposed method.

3. The proposed Self-reflective Contrastive Learning is interesting and effective.

**Questions For Authors:**

1. Could the proposed method be evaluated on more general datasets, e.g., Imagenet?

2. Could the proposed method be compared with other RAG-based methods?

3. Could a detailed computational overhead analysis be added?

4. Could more failure case analysis be discussed?

**Relation To Broader Scientific Literature:**

Some previous work used image retrieval to improve image generation. For example, [1], [2], [3], and some more recent ones.

[1] knn-diffusion: Image generation via large-scale retrieval. ICLR 2022.

[2] Retrieval-augmented diffusion models. NeurIPS 2022.

[3] Retrieval-augmented text-to-image generator. ICLR 2022.

**Theoretical Claims:**

Yes.

---

> ### Author Rebuttal · Authors · 2025-03-31
>
> **We sincerely thank *Reviewer HZx9*  for the constructive comments and insightful suggestions.**
>
> # Evaluation on More General Datasets
>
> Thanks for the reviewer's suggestion. We add the experiments with the ImageNet dataset, please check the results in the response for ***Reviewer 5Kbn***. The results demonstrate that our RealRAG showcases strong performance on the general datasets.
>
> # Compared with Other RAG-based Methods
>
> The reference mentioned by the reviewer in the **Essential References Not Discussed** section, referred to here as RDMs, and our RealRAG both utilize retrieval-augmented techniques in text-to-image generation. However, the research problems and objectives we aim to solve are entirely different.
>
> * **Research Purpose of RDMs**: They use retrieval-augmented techniques to ***train or fine-tune*** a diffusion model and achieve ***out-of-distribution (OOD) image generation by switching databases***.
>
> * **Research Purpose of RealRAG**: RealRAG uses retrieval-augmented techniques to ***mitigate hallucinations*** when generating fine-grained objects and to enhance the ability to generate unseen novel objects, ***without requiring the training of generative models.***
>
> **Out-of-Distribution Objects ≠ Unseen Novel Objects**:
>   * **Out-of-distribution objects**: In the RDMs work mentioned by the reviewer, out-of-distribution is defined as data from outside the domain of the training dataset, such as "**Angry Bird**"(Knn-diffusion[1]) These objects existed during the generative model's training but were not part of the training set. As a result, the foundational generative model cannot generate them. RDMs use retrieval-augmented techniques, employing the CLIP model's similarity calculation ability to provide relevant references to the generative model, thereby solving the OOD problem.
>   * **Unseen novel objects**: These refer to objects that appear after the generative models and retrieval models are trained. The generative model cannot generate these objects, ***and the retrieval model can't easily retrieve relevant references via only similarity-based retrieval***. This is a much more challenging issue.
>
> For **OOD object generation**, existing state-of-the-art text-to-image generators can generate images from almost all domains, as shown in the [link](https://realrag.github.io/AngryBird/), and the original FLUX model can generate "**Angry Bird**" very well, outperforming the retrieval-augmented KNN-diffusion[1]. However, for **unseen novel objects**, such as the "**cybertruck**" (shown in **Fig. 5** of the main paper), FLUX still cannot generate an accurate cybertruck. Therefore, the focus of this paper is on ***solving the problem of generating unseen novel objects.***
>
> Finally, although RDM is not part of the same type of research as our RealRAG, we still **provide a comparison of their performance in the following table.**
>
> | Models          | CLIP-I | CLIP-T |  FID  |
> | --------------- | :----: | :----: | :---: |
> | RDM[2]          | 59.82  | 12.37  | 69.20 |
> | CLIP-similarity | 61.72  | 14.52  | 54.04 |
> | RealRAG         | **62.81**  | **14.46**  | **52.28** |
>
> Our RealRAG shows significant performance gain, and we also show more visual comparison between the baseline and our RealRAG here [link](https://realrag.github.io/RealRAG_Rebuttal/).
>
> # Computational Overhead Analysis
>
> We present the retrieval time cost and the performance gain in the table below. The table shows that the average percentage increase in reasoning time is much lower than the percentage increase in performance. Our approach achieves significant performance gains over a limited increase in inference time.
>
> | Model | Original Time Cost | RAG Time Cost | Delay (%) | AVG FID | AVG RAG FID | Gain (%) | Gain / Delay |
> | :---- | :----------------: | :-----------: | :-------: | :-----: | :---------: | :------: | :----------: |
> | Emu   |        8.96        |     9.32      |   **3.57**    |  73.27  |    59.81    |  **18.37**   |     ***+5.15***     |
> | SDXL  |        5.94        |     6.30      |   **6.06**    |  55.48  |    51.22    |   **7.68**   |     ***+1.27***     |
> | FLUX  |       13.42        |     13.78     |   **2.68**    |  53.37  |    49.28    |   **7.66**   |     ***+2.86***     |
>
> # Failure Cases
>
> This work, like original FLUX and Stable Diffusion, has not been designed to target multi-object generation, therefore, RealRAG has some limitations in multi-object generation. Here we show some failure cases of multi-object generation [link](https://realrag.github.io/FailureCase/). ***It is our future work to implement multi-object generation***.

---

> > ### Comment · Reviewer_HZx9 · 2025-04-04
> >
> > Thank you for the detailed and thoughtful response. I appreciate the clarifications provided. I will maintain my rating as weak accept.

---

> > > ### Author Response · Authors · 2025-04-04
> > >
> > > Thank you sincerely for your time, expertise, and constructive engagement throughout the review process. We deeply appreciate your recognition of our work, as well as your valuable feedback that has significantly improved the quality of this work.

---

### Official Review · Reviewer_5Kbn · 2025-03-15

**Overall Recommendation:** 3

**Summary:**

The paper introduces a novel retrieval-augmented generation (RAG) framework aimed at improving text-to-image generative models. Traditional generative models suffer from hallucinations and distortions when generating fine-grained or novel real-world objects due to their fixed training datasets. RealRAG overcomes this limitation by integrating external real-world images through a self-reflective contrastive learning approach. This technique ensures that the retrieved images supplement missing knowledge rather than merely matching text prompts based on similarity. RealRAG is adaptable to various generative architectures, including diffusion and auto-regressive models, yielding significant performance improvements. The method demonstrates superior realism in generating both fine-grained and unseen objects, outperforming existing retrieval-based models while maintaining modular compatibility across different generative approaches.

**Claims And Evidence:**

Yes

**Essential References Not Discussed:**

No.

**Experimental Designs Or Analyses:**

Yes, the experiment design is reasonable and relatively sufficient.

**Methods And Evaluation Criteria:**

Yes

**Other Comments Or Suggestions:**

No.

**Other Strengths And Weaknesses:**

Strengths:
1. The idea for using contrastive learning to retrieve images as supplementary knowledge for pre-trained generative models is novel.
2. The writing is easy to follow, and the figures help readers to understand the paper.
3. Experimental results show that the proposed method significantly improves the generation ability on various datasets with fine-grained classes.

Weaknesses:
1. Although the idea is novel, I am wondering why not just search relative images from the internet rather than retrieve them from the image pool. It should also work well.
2. The number of classes and dataset size is relatively small in the experiments part. Larger datasets with more complicated and challenging classes need to be evaluated.
3. The image as a reference part is not clear to me. Existing generative models (including SD based and AR based) only support text condition as input, how would you use image as additional reference? By cross-attention or initialize the latent noise with the reference images? This is a very important part, and you need to explain it clearly in the main paper for readers to understand.

**Questions For Authors:**

See weaknesses.

**Relation To Broader Scientific Literature:**

es, related to real-world applications for enhancing generative models' ability to generate new concepts.

**Theoretical Claims:**

No theoretical proof in this paper

---

> ### Author Rebuttal · Authors · 2025-03-31
>
> **We sincerely thank *Reviewer 5Kbn* for the constructive comments on our work. We are very grateful to the reviewer for recognising the novelty of our idea and the richness and rationality of our experiments.**
>
> # About the Database
>
> ***We are sorry for the misunderstanding***. For the fine-grained object generation setting, we use the Real-object-based Database; For the unseen novel object generation setting, as shown on line 042, we use the images from the Internet (**Google Images**), which include the **novel objects**.
>
> We show the results of using Internet data in the **Fig. 5 in the main paper**.
>
> # Evaluation on ImageNet
>
> Thanks for the reviewer's suggestion. We add the experiments with the ImageNet.
>
> | Model        |          | ImageNet Val |          |
> | ------------ | :------: | :----------: | :------: |
> |              |  CLIP-I  |    CLIP-T    |   FID    |
> | Emu          |  53.55   |    14.18     |  25.56   |
> | Emu W. Ours  |  55.48   |    16.21     |  23.51   |
> | **Gain**     | **1.93** |   **2.03**   | **2.05** |
> | SDXL         |  53.74   |    14.53     |  17.57   |
> | SDXL W. Ours |  57.35   |    17.20     |  12.81   |
> | **Gain**     | **3.61** |   **2.67**   | **4.76** |
> | FLUX         |  54.41   |    14.77     |  16.84   |
> | FLUX W. Ours |  56.73   |    15.82     |  14.56   |
> | **Gain**     | **2.31** |   **1.05**   | **2.28** |
>
> Our RealRAG demonstrates significant performance gains for the SoTA t2i generative models on the ImageNet dataset. Furthermore, our insight is to **reduce hallucinations of the fine-grained object generation** (eg, various types of cars) and **the classes in ImageNet are coarse-grained** (eg, cars and planes). Therefore, the selected datasets (eg, Stanford Car) are more suitable for the evaluation.
>
> # More clarification about Image-conditioned generation
>
> *So sorry for the lack of a relevant background introduction.* Several methods have been proposed for inputting images into diffusion models. These include: 1) **Embedding-based approaches**, where image embeddings from CLIP are concatenated with timestamp embeddings in the UNet of diffusion models [Ref A, B]; 2) **Input Layer Concatenation-based approaches**, in which the latent representation of the input image is concatenated to the UNet's input, changing its input dimension from 4 to 8, with the additional input layer initialized to zero [Ref C]; 3) **ControlNet-based approaches**, where ControlNet [Ref D] introduces a branch to stable diffusion, enabling the inclusion of additional inputs; and 4) **Noise-based approaches**, which involve image editing methods [Ref E] that add noise to the input image, followed by denoising.
>
> We use the version of our **Reflective Retriever + FLUX** as an example. First, we retrieve and sort the closest images. Next, we input the selected images into the FLUX’s ControlNet branch to control specific elements during the image synthesis process. The FLUX version is **black-forest-labs/FLUX.1-Canny-dev**, which allows for inputting a referenced image to guide the generation process. For further details, please refer to the code provided in the supplementary materials.
>
> In general, equipped with our Reflective Retriever, we can significantly enhance the prompt-image consistency and feasibility.
>
> [Ref A] https://huggingface.co/lambdalabs/sd-image-variations-diffusers.
>
> [Ref B] https://huggingface.co/stabilityai/stable-diffusion-2-1-unclip.
>
> [Ref C] Zero-1-to-3: Zero-shot one image to 3d object.
>
> [Ref D] Adding Conditional Control to Text-to-Image Diffusion Models.
>
> [Ref E] Sdedit: Guided image synthesis and editing with stochastic differential equations.

---

> > ### Comment · Reviewer_5Kbn · 2025-04-04
> >
> > Thanks for the rebuttal. The additional results look good to me. Since my original score is weak accept, my score will remain the same. I hope the authors can revise the paper as I suggested in the final version if accepted.

---

> > > ### Author Response · Authors · 2025-04-04
> > >
> > > Thanks for taking the time to review our paper and for providing insightful feedback. We highly value the insights and suggestions, which we believe will significantly enhance the quality of our work. **We promise to revise the paper based on the comments.** *We sincerely appreciate your recognition and support of our work*.

---

### Official Review · Reviewer_vQw4 · 2025-03-17

**Overall Recommendation:** 3

**Summary:**

The paper introduces RealRAG, a retrieval-augmented generation framework designed to enhance text-to-image models by addressing their inherent knowledge limitations. The main idea is to retrieve and integrate real-world images to supplement the generator's missing knowledge. The key innovation of RealRAG lies in its reflective retriever, trained using self-reflective contrastive learning. This method ensures that retrieved images effectively compensate for gaps in the model's knowledge. Experimental results comparing on fine-grained object generation and unseen novel object generation demonstrate its effectiveness.

##update after rebuttal
The authors have addressed most of my concerns satisfactorily, I am therefore updating my recommendation to a weak accept.

**Claims And Evidence:**

1. The paper claims to present the first real-object-based retrieval-augmented generation (RAG) framework. However, the term real-object-based appears to be primarily reflected in the fact that the retrieval dataset is constructed using real objects. This concept, however, is not novel, as prior works have also employed real-object-based datasets in a similar retrieval-augmented manner. For instance, KNN-Diffusion [1] utilizes the MS-COCO dataset, while RDM [2] leverages the ImageNet dataset to enhance generative processes. Given these precedents, this claim requires further clarification.

2. Furthermore, the paper claims to be the first to establish a unified RAG framework applicable to all major categories of text-to-image generative models, including diffusion models and autoregressive models. However, a similar direction has already been explored in RDM, which proposes a RAG system capable of being integrated with multiple likelihood-based generative methods, also including both diffusion models and autoregressive models. In light of this, the assertion of novelty should be carefully contextualized with respect to prior works.

[1]Shelly Sheynin, Oron Ashual, Adam Polyak, Uriel Singer, Oran Gafni, Eliya Nachmani,
Yaniv Taigman. "Knn-diffusion: Image generation via large-scale retrieval."
International Conference on Learning Representations, 2023.

[2] Andreas Blattmann, Robin Rombach, Kaan Oktay, Jonas Müller, Björn Ommer.
"Retrieval-augmented diffusion models." Advances in Neural Information Processing
Systems, 2022.

**Essential References Not Discussed:**

This paper focuses on the field of Retrieval-Augmented Image Generation; however, the discussion of related work lacks a comprehensive review of prior retrieval-augmented generative models.

[1]Robin Rombach, Andreas Blattmann, Björn Ommer. "Text-guided synthesis of artistic
images with retrieval-augmented diffusion models." arXiv preprint, 2022.

[2] Andreas Blattmann, Robin Rombach, Kaan Oktay, Jonas Müller, Björn Ommer.
"Retrieval-augmented diffusion models." Advances in Neural Information Processing
Systems, 2022.

[3]Shelly Sheynin, Oron Ashual, Adam Polyak, Uriel Singer, Oran Gafni, Eliya Nachmani,
Yaniv Taigman. "Knn-diffusion: Image generation via large-scale retrieval."
International Conference on Learning Representations, 2023.

[4] Wenhu Chen, Hexiang Hu, Chitwan Saharia, William W. Cohen. "Re-imagen:
Retrieval-augmented text-to-image generator." arXiv preprint arXiv:2209.14491 (2022)

[5] Huaying Yuan, Ziliang Zhao, Shuting Wang, Shitao Xiao, Minheng Ni,
Zheng Liu, and Zhicheng Dou. Finerag: Fine-grained retrieval-augmented
text-to-image generation. In Proceedings of the 31st International Confer-
ence on Computational Linguistics, pages 11196–11205, 2025.

**Experimental Designs Or Analyses:**

1. The experimental comparison should be conducted against other retrieval-augmented image generation methods, as specified in the "Essential References Not Discussed" section. A comprehensive comparison with these methods will provide a clearer evaluation of the proposed approach's effectiveness and highlight its advantages and limitations in relation to existing techniques.

2. The retrieval database used in the unseen novel object generation scenario requires further clarification. Specifically, it is important to confirm whether the database still consists of ImageNet, Stanford Cars, Stanford Dogs, and Oxford Flowers. Additionally, if these datasets are indeed used, it would be better to illustrate which images are retrieved in this unseen novel object generation case. Would the retrieved images provide useful informations.

3. The number of image sets used for human evaluation appears to be relatively small, with only four sets being considered. This limited sample size may not be sufficient for a robust quantitative analysis, potentially affecting the statistical significance of the results.

4. The type of embedding used to represent the constructed representation spaces in both the standard retrieval-augmented generation (RAG) approach and RealRAG in t-SNE visualization should be explicitly stated.

**Methods And Evaluation Criteria:**

The proposed method utilizes several benchmark datasets, including ImageNet, Stanford Cars, Stanford Dogs, and Oxford Flowers, as the retrieval database. However, the evaluation of the proposed RealRAG framework is conducted exclusively on three fine-grained real-world image datasets: Stanford Cars, Stanford Dogs, and Oxford Flowers.

This raises two key concerns.

1. Given that ImageNet is included in the retrieval database, it is unclear why the evaluation does not extend to general image generation using ImageNet. Such an evaluation could provide insights into the method’s performance on broader image retrieval and generation tasks.

2. The inclusion of ImageNet as part of the retrieval database introduces an important trade-off: while a larger retrieval database may enhance retrieval diversity and coverage, it also increases retrieval time complexity. Therefore, it is crucial to analyze the computational impact of incorporating ImageNet into the retrieval database and determine whether the associated increase in retrieval time justifies the potential performance improvements.

**Other Comments Or Suggestions:**

1. The illustrations in Figure 1, specifically images (1) and (2), appear to be misarranged. Their correspondence with the description provided in the Introduction section is inconsistent, leading to potential confusion for readers.

2. In Section 2.1, the reference to the large-scale text-image paired dataset incorrectly cites "LIANG 5B" instead of "LAION 5B." The correct dataset name should be "LAION 5B," a widely recognized and publicly available resource.

3. In the Experiments section, specifically on line 271 of page 5, there is an empty set of parentheses following the mention of the CLIP model, indicating a missing reference.

4. In the qualitative results section, on line 308 of page 6, the discussion regarding the comparison with the AR model does not correspond to the correct text prompt.

**Other Strengths And Weaknesses:**

The strengths and weaknesses have been presented above.

**Questions For Authors:**

1. It would be important for authors to explicitly clarify how their approach differs from and improves upon these prior works. If they acknowledge that these claims require revision, they should provide a more precise positioning of their contributions and add a more comprehensive discussion.

2. Why was ImageNet excluded from the evaluation? Including it could provide a broader perspective on the method’s performance.

3. The retrieval database includes ImageNet, which increases retrieval diversity but also adds computational cost. Could the authors provide a quantitative analysis of the retrieval time impact?

4. The experimental comparison should include other retrieval-augmented image generation methods such as those listed in the "Essential References Not Discussed" section.

5. In the unseen novel object generation scenario, does the retrieval database still include ImageNet, Stanford Cars, Stanford Dogs, and Oxford Flowers? If so, could the authors provide examples of retrieved images and explain whether they provide meaningful information for generating unseen objects? This would help clarify whether the retrieval process effectively supports novel object generation.

**Relation To Broader Scientific Literature:**

The idea of exploring the generator's missing knowledge for retrieval-augmented generation task might be related to the broader scientific literature.

**Theoretical Claims:**

This paper does not present any formal proofs or theoretical claims.

---

> ### Author Rebuttal · Authors · 2025-03-31
>
> **We sincerely thank *Reviewer vQw4*  for the constructive comments on our work. We promise to revise the paper based on the comments. We will cite and discuss all the papers in the "Essential References Not Discussed" in the final version**
>
> # About the "first work" claim and the novelty.
> The reference mentioned by the reviewer in the **Essential References Not Discussed** section, referred to here as RDMs, and our RealRAG both utilize retrieval-augmented techniques in text-to-image generation. However, the research problems and purposes we aim to solve are entirely different.
> * **Research Purpose of RDMs**: They use retrieval-augmented techniques to ***train or fine-tune*** a diffusion model and achieve ***out-of-distribution (OOD) image generation*** by switching databases.
> * **Research Purpose of RealRAG**: RealRAG uses retrieval-augmented techniques to ***mitigate hallucinations*** when generating fine-grained objects and to enhance the ability to generate ***unseen novel objects***, ***without any training*** of generative models.
>
> There are **three key points** to pay attention to:
> 1. **Out-of-Distribution Objects ≠ Unseen Novel Objects**:
>    - **Out-of-distribution objects**: In the RDMs work mentioned by the reviewer, out-of-distribution is defined as data from outside the domain of the training dataset, such as **"Angry Bird."** In **Knn-diffusion**[3]. These objects existed during the generative model's training but were not part of the training set. As a result, the foundational generative model cannot generate them. RDMs use retrieval-augmented techniques, employing the CLIP model's similarity calculation ability to provide relevant references to the generative model, thereby solving the OOD problem.
>    - **Unseen novel objects**: These refer to objects that appear after the generative models and retrieval models are trained. The generative model cannot generate these objects, **and the retrieval model can't easily retrieve relevant references by similarity**. This is a much more challenging issue.
>
>    For OOD object generation, existing state-of-the-art text-to-image generators can generate images from almost all domains, as shown in the [link](https://realrag.github.io/AngryBird/), and the original FLUX model can generate **"Angry Bird"** very well, outperforming the retrieval-augmented KNN-diffusion. However, for unseen novel objects, such as the **"cybertruck"** (shown in **Fig. 5** of the main paper), FLUX still can't generate an accurate cybertruck. Therefore, the focus of this paper is on ***solving the problem of generating unseen novel objects.***
> 2. **Hallucination Issue When Generating Fine-Grained Objects**:
>
>    Existing SoTA t2i models are pre-trained on large-scale text-image paired datasets. As a result, they tend to produce hallucinations, such as inaccuracy or unrealistic features, when generating fine-grained objects. This is a problem inherent to large generative models.
> 3. **Application Value**:
>
>    For existing commercial text-to-image generative models, training them is cost-prohibitive. Therefore, a pipeline is needed to integrate real-time updated data from the internet into the generative model **without any retraining**, enabling the generation of **unseen novel objects**. On the other hand, in specific application scenarios such as advertising or poster creation, users need generated images that meet their design requirements, while also ensuring that **products (fine-grained objects) within the image remain realistic**. This requires generative models to have the ability to generate **both open-domain and fine-grained objects**. Therefore, RealRAG focuses on reducing hallucinations in large t2i generators through RAG technology, enabling open-domain generators to generate specific fine-grained objects.
> # Evaluation on ImageNet.
> Thanks for the reviewer's suggestion. Please check the results in the rebuttal for ***Reviewer 5Kbn***.
> # More Baselines
> Please check the results in the response for ***Reviewer HZx9***.
> We also show more visual comparison here [link](https://realrag.github.io/RealRAG_Rebuttal/).
> # About the Database
> We are sorry for the misunderstanding. For the fine-grained object generation setting, we use the Real-object-based Database. For the unseen novel object generation setting, as shown on line 042, we use the images from the Internet (Google Images), which include the novel objects.
> # More User Study
> Thanks for your suggestions, we **try our best to add more cases and invite more participants** for the user study. Finally, we extant the scale of the user study to **50 participants and 20 cases**. The results can be found at the [link](https://realrag.github.io/UserStudy/).
> # About **the trade-off between performance and the inference time**.
> We present the comparison results in the response for ***Reviewer HZx9***. The table shows that the average percentage increase in reasoning time is much lower than the percentage increase in performance.

---

### Decision · Program_Chairs · 2025-05-01

**Decision:**

Accept (poster)

**Comment:**

RealRAG introduces a novel framework for t2i generation to improve the fixed-knoledge generation models. It introduces self-reflective contrastive learning to train a reflective retriever. This retriever then identifies and retrieves “reflective negatives” from an external database to compensate for missing visual details not included in the generated image. Experiments show that RealRAG significantly improves synthesized images' realism and fine-grained accuracy across various architectures.

The paper received 3 weak accepts, with all reviewers recognizing its novelty and technical contributions and expressing satisfaction with the rebuttal. Based on these positive assessments, the area chairs recommended acceptance.